# Comparison of Immune Response of *Litopenaeus vannamei* Shrimp Naturally Infected with *Vibrio* Species, and after Being Fed with Florfenicol

Medhat S. Shakweer [1], Gehad E. Elshopakey [2], Abdelwahab A. Abdelwarith [3], Elsayed M. Younis [3], Simon John Davies [4] and Samia Elbahnaswy [1,*]

1 Department of Internal Medicine, Infectious and Fish Diseases, Faculty of Veterinary Medicine, Mansoura University, Mansoura 35516, Egypt
2 Department of Clinical Pathology, Faculty of Veterinary Medicine, Mansoura University, Mansoura 35516, Egypt
3 Department of Zoology, College of Science, King Saud University, P.O. Box 2455, Riyadh 11451, Saudi Arabia
4 College of Science and Engineering, University of Galway, H91 TK33 Galway, Ireland
* Correspondence: samiaahmed@mans.edu.eg; Tel.: +20-1098668747; Fax: +20-50-2239963

**Abstract:** The outbreaks caused by *Vibrio* spp. are a notable threat to the potential growth of the economy of penaeid culture, which is still controlled by the administration of antibiotics. At first, the infected group was subjected to phenotypic bacteriological examination with subsequent molecular identification via 16S rRNA gene sequencing, which confirmed four strains of *Vibrio* spp., *V. atlanticus*, *V. natriegens*, *V. alginolyticus*, and *V. harveyi*, from moribund-infected shrimp during mortality events in an Egyptian hatchery. To better understand the defense mechanism of the most effective antibiotic against *Vibrio* strains, the immune responses were compared and evaluated in infected *Litopenaeus vannamei* broodstock after being fed 5 mg kg$^{-1}$ of florfenicol antibiotic, which was first determined through in vitro antibiogram tests. Therefore, our study aimed to determine the immune response of *L. vannamei* during *Vibrio* spp. infection in Egyptian hatcheries and after antibiotic medication. The parameters assessed were the total and differential hemocyte count (THC), granular cells (GC), semi-granular cells (SGC), and hyaline cells (HC). As well as the metabolic and immune enzymes: alanine aminotransferases (ALT), aspartate aminotransferases (AST), alkaline phosphatase (ALP), acid phosphatase (ACP), and lysozyme activity; an antioxidant index, such as superoxide dismutase (SOD) and glutathione (GSH); a phagocytic assay; changes in reactive oxygen species (ROS); and bactericidal activity in the hemolymph of the control, infected, and treated groups. Further evaluation of the mRNA expression levels of the prophenoloxidase (*LvproPO*), toll-like receptor 1 (*LvToll1*), and haemocyanin (*LvHc*) genes were performed in the hepatopancreas of the same groups. A significant drop in the THC, GC, SGC, and HC counts, as well as lysozyme and bactericidal activities, phagocytic assay, ROS, SOD, and GSH index, were represented in infected shrimp compared to control shrimp; however, a marked increase in the activity of ALT, AST, ALP, and ACP was observed. These activities were significantly restored in the treated shrimp compared to the infected shrimp. Nevertheless, no significant changes were noted in the transcriptional levels of the *LvproPO* and *LvToll1* genes in the treated shrimp when compared to the infected shrimp; however, a significant suppression of the *LvHc* gene was noted. Our study aimed to determine the immune response of *L. vannamei* during *Vibrio* spp. infection in Egyptian hatcheries and after antibiotic medication. We concluded that florfenicol in medicated feed could be effective in controlling vibriosis and ameliorating the immune response of shrimp.

**Keywords:** *Litopenaeus vannamei*; vibriosis; 16S rRNA sequencing; hematology; immune response; gene expression





## 1. Introduction

Shrimp farming is recognized as a major economic source in the aquaculture sector for its high protein content for human nutrition [1]. Pacific white-leg shrimp, *Litopenaeus vannamei*, has become one of the main economically cultivated species in Latin America and Southeast Asia [2]. Egypt produces marine shrimp in semi-intensive culture systems with an annual production of more than 7000 tons; however, the majority of shrimp hatcheries still depend on wild broodstock to produce the postlarvae [3,4]. Infectious microbes comprise one of the leading constraining factors in *L. vannamei* mariculture industries, causing global economic losses of more than 1 billion USD, especially from *Vibrio* spp. [5,6]. The most serious bacterial pathogens are the gram-negative *Vibrio* spp. in the family *Vibrionaceae*, which include more than 100 species, such as *V. anguillarum*, *V. alginolyticus*, *V. parahaemolyticus*, *V. harveyi*, *V. penaeicida*, and *V. campbelli* [7,8]. Vibriosis is the most important and challenging disease in penaeid shrimp hatcheries, resulting in low survival rates of commercial postlarvae and broodstock shrimp culture worldwide [9]. Moreover, the widespread dissemination of *Vibrio* spp. in shrimp culture ecosystems is related to their rapid multiplication rates and ability to acclimate to environmental alterations [10]. Most of the virulence factors of *Vibrio* spp. involve hemolysins, cytotoxins, and iron-acquisition systems [11]. In particular, the bioluminescent marine bacterium *V. harveyi*, which causes penaeid bacterial septicemia, has been reported to induce 100% mortality in the early larval stages of shrimp, with enormous production losses in the shrimp industry [2,12]. A variety of virulence factors, such as toxin-A production, adhesion factors, extracellular polysaccharides, biofilm formation, lytic enzymes, siderophores, type III secretion systems, and bacteriophages, induce *V. harveyi* infections [13]. In addition, *V. alginolyticus* is a recurrent pathogen in shrimp hatcheries worldwide [14]. In the past three decades, an array of phenotypic and molecular methods has been applied for *Vibrio* spp. identification. However, traditional phenotypic characterization methods of *Vibrio* spp. are time-consuming and have restricted discriminatory power because *Vibrio* spp. have highly similar phenotypes, particularly *V. alginolyticus*, which is biochemically very similar to *V. parahaemolyticus* [15]. Advanced genomic techniques, such as DNA-sequence-based identification and 16S rRNA, have increased the diagnostic accuracy of *Vibrio* spp. [8]. The control of bacterial pathogens triggered by *Vibrio* primarily depends on antibiotic usage in hatcheries and shrimp farms [16,17]. However, studies have confirmed that overuse of antibiotics in shrimp farming has resulted in the evolution of several types of resistant *Vibrio* spp. [18], and they could also destroy microbially mature systems combined with their ineffectiveness against luminescent *V. harveyi* [19]. Florfenicol (FLO), one of the most widely used antibacterial drugs, penetrates the cells via facilitated transport by blocking the union site of the 50S ribosomal subunit. This antibiotic is effective against chloramphenicol-resistant bacteria but lacks the functional set, which is specific to human toxicity [20,21]. Consequently, it has been licensed by the Food and Drug Administration (FDA) in the USA [22] and in many other countries for the control of mortality in farmed catfish and salmonids infected with columnaris disease [23], nile tilapia in Brazil [24], and cod *Gadus morhua* against *V. anguillarum* [25]. Relevant studies have evaluated the potential impact of florfenicol in controlling shrimp mortality induced by *Vibrio* spp. [26–28].

Shrimp have an innate immune response mediated by hemocytes, which includes cellular and humoral reactions [29,30]. Cell-mediated immunity includes phagocytosis by hyaline cells (HCs) and encapsulation and nodule formation by granular cells (GCs) and semi-granular cells (SGCs) [31,32]. The primary humoral response involves the secretion of pattern-recognition proteins and a variety of humoral effectors, like the prophenoloxidase (proPO) system, as well as antimicrobial peptides (AMPs), and clotting proteins to degrade the invading pathogens. Regarding antimicrobial defense, the production of both reactive oxygen intermediates (ROIs) and toxic intermediates quinones through the proPO system is a potent mechanism in killing the pathogens in crustaceans [33,34]. Evaluation of shrimp immunity during a bacterial infection can provide important clues to understanding vulnerability to disease in shrimp and subsequently identify immunomarkers, which could

be valuable in the diagnosis and management of outbreaks of disease in penaeids. Therefore, the goal of this work was to determine, through in vitro analyses, the most effective antibiotic against pathogenic *Vibrio* spp. isolated from *L. vannamei* from a hatchery suffering high winter mortalities in the Dibah Triangle Zone (DTZ), Damietta-Port-Said Province, Egypt. We performed a molecular characterization of the isolated *Vibrio* spp. using *Vibrio* 16S rRNA as a target marker, followed by sequencing and phylogenetic analyses. In addition, the antibiotic effectiveness of florfenicol was investigated by evaluating the immune response of shrimp and the immune-related gene expression in response to natural infection by *Vibrio* spp. and after florfenicol administration.

## 2. Materials and Methods

### 2.1. Collection of Shrimp Samples from Study Area

Diseased broodstock of *L. vannamei* (body weight 45.68 g $\pm$ 0.18 g) were randomly collected from three large raceways (100 shrimp per raceway, 3 m $\times$ 8 m $\times$ 1 m) set in a shrimp hatchery in DTZ, Damietta-Port-Said Province, Egypt (Figure 1), during a mortality event in the winter season from January to April, 2021. These raceways (El-Ekhlas shrimp hatchery, DTZ, Damietta-Port-Said Province, Egypt) were classified as the 'infected' group where moribund shrimp were observed to have high mortality rates (55%) and displayed lethargic swimming behavior and congestion of almost all body appendages and telson with brown and hemorrhagic discoloration of the hepatopancreas (HP). The feeding regime was 5% of the total body weight of a basal commercial feed containing 40% protein (Skretting, Nutreco Company, Amersfoort, Netherlands). Water quality parameters, such as ammonia (mg L$^{-1}$) via the LaMotte$^{®}$Aquaponics Test Kit (code no. 3637) (LaMotte Company, Chestertown, MD, USA), temperature ($^{\circ}$C) using a water thermometer (Yellow Springs Comp., Yellow Springs, OH, USA), dissolved oxygen (mg L$^{-1}$) using an oximeter model YK-22DO, and salinity (‰) using a salinometer alongside the pH value, were monitored daily. Shrimp samples (10 random moribund shrimp/raceway) were transported from the hatchery to the research facilities of the Laboratory of Fish Diseases and Management, Veterinary Medicine, Mansoura University, with a maximum time interval of 2 h. Polyethylene bags filled with marine water (one third) and oxygen (two thirds) prior to sealing were used for the transportation of shrimp. Consequently, sampling of the diseased shrimp was performed within two days of the disease outbreak. At first, hemolymph of 100 μL (n = 6) was collected from the ventral sinus of *L. vannamei* samples via a syringe prefilled with 500 μL of an anticoagulant buffer (trisodium citrate 30 mM, sodium chloride 0.34 M, EDTA 10 mM, pH 7.5, osmolality 780 mOsm kg L$^{-1}$) for the evaluation of total hemocyte count, phagocytosis, and respiratory burst activity. Another hemolymph sample (100 μL) was individually collected from each shrimp in different raceways, stored at 4 $^{\circ}$C for 12 h, and then centrifuged at 4000 rpm for 10 min. The supernatant was collected and kept at $-80$ $^{\circ}$C until further analyses of enzyme activities, antioxidant index, and immunological parameters. Simultaneously, shrimp hepatopancreas (HP) from the three groups was aseptically tested for bacterial isolation and then carefully dissected (vertical cut) into two longitudinal sections, and one section from each sample was immediately fixed in RNA*later*$^{®}$ (Sigma-Aldrich, Inc., St. Louis, MO 68178, USA) and stored at $-80$ $^{\circ}$C for gene expression analysis.

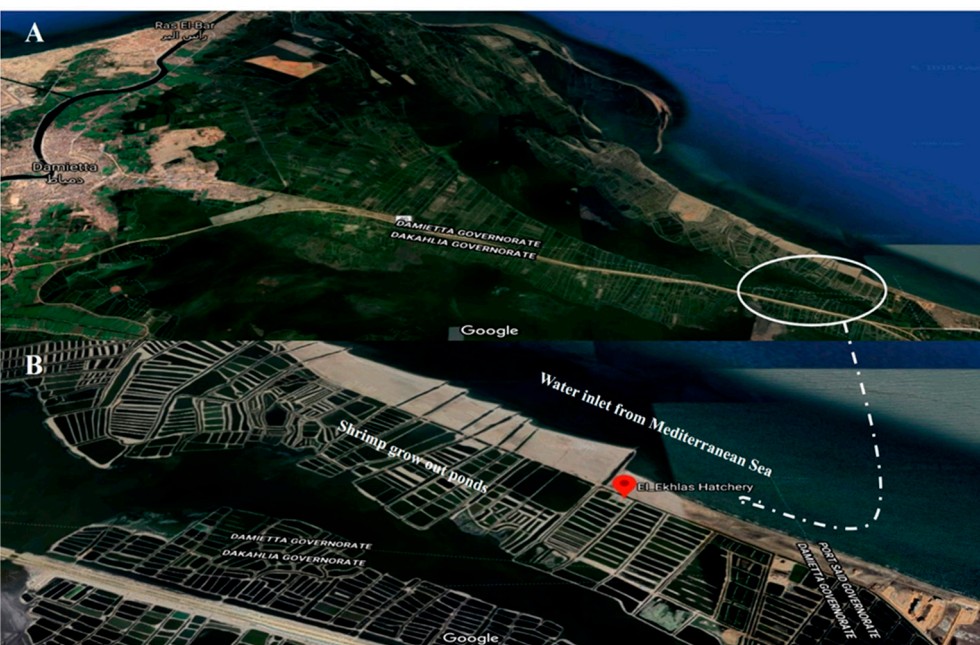

**Figure 1.** (**A**): Satellite map from Google Earth (https://earth.google.com/web/search/El-Ekhlas+shrimp+hatchery+in+damietta/ (accessed on 3 January 2023)) shows the Dibah Triangle Zone (DTZ), Damietta-Port-Said Province, Egypt. (**B**): El-Ekhlas shrimp hatchery (sampling site) surrounded by shrimp growout ponds.

### 2.2. Isolation and Identification of Vibrio spp. Using 16S Ribosomal RNA Sequence Analysis

The surface of the diseased shrimp was disinfected with 75% alcohol spray (EasyCare, El-Giza, Egypt), and the sampled hepatopancreas was collected under aseptic conditions and streaked directly on thiosulfate-citrate-bile-salts-sucrose agar (TCBS, Difco™, Becton and Dickinson, New York, NY, USA), which was incubated for 18–24 h at 28 ± 2 °C, and then purified colonies were restreaked on TCBS to obtain pure bacterial colonies.

Bacterial isolates were initially identified by biochemical characterization as per the criteria in Bergey's Manual of Determinative Bacteriology [35]. Further presumptive characterization of the retrieved isolates was achieved using a commercial miniaturized API®20NE system (BioMérieux, Inc., Durham, NC, USA) according to the manufacturer's instructions. The isolates were identified according to previously described diagnostic schemes [36,37]. The purified strains were stored in brain heart infusion broth 15% (CM1135, Oxoid, Basingstoke, UK) (vol/vol) glycerol (Sigma-Aldrich, Inc., St. Louis, MO, USA) at 20 °C.

Molecular identification of retrieved *Vibrio* spp. was performed using qualitative PCR. Identified bacterial colonies were cultured on tryptic soya agar +2% NaCl and then were picked up for DNA extraction using a QIAamp DNA Mini kits (Qiagen, Redwood, CA, USA) with some modifications according to the manufacturer's protocol. Briefly, 200 µL of the sample suspension was incubated with 10 µL of proteinase K and 200 µL of lysis buffer at 56 °C for 10 min. After incubation, 200 µL of 100% ethanol was added to the lysate, which was then washed and centrifuged. The nucleic acid was then eluted with 100 µL of elution buffer. Two primer sets for *Vibrio* 16S rRNA (5′-CGGTGAAATGCGTAGAGAT-3′) and (5′-TTACTAGCGATTCCGAGTTC-3′) were used to perform PCR, which amplified a 663-bp fragment [38]. PCR amplification was performed in a 25-µL reaction mixture containing 12.5 µL of Emerald Amp Max PCR Master Mix (Takara, Tokyo, Japan), 1 µL of each primer (20 pmol; Metabion, Planegg, Germany), 4.5 µL of distilled water, and 6 µL of DNA template. *V. parahaemolyticus* (ATCC® 17802™) and *Escherichia coli* (ATCC®25922™) were used as positive and negative controls, respectively. The reaction proceeded in an Applied Biosystems 2720 thermal cycler (Applied Biosystems, Foster City, CA, USA). The thermal profile of PCR consisted of an initial denaturation step at 94 °C for 5 min, followed

by 35 cycles of 94 °C for 30 s, 56 °C for 45 s, and 72 °C for 45 s, and a final extension step at 72 °C for 10 min. Next, 15 μL of PCR products were analyzed by agarose (1.5%) gel electrophoresis (AppliChem GmbH, Darmstadt, Germany) using gradients of 5 V cm$^{-1}$, and a 100-bp DNA ladder (Fermentas, Thermo Fisher Scientific, Waltham, MA, USA) was used to determine the fragment sizes. The bands were photographed using a gel documentation system (Alpha Innotech, Biometra, San Leonardo, CA, USA).

The QIAquick PCR product extraction kits (Qiagen, Valencia, Spain) were used to purify PCR products. Subsequently, Bigdye Terminator V3.1 cycle sequencing kits (Perkin Elmer Applied Biosystems, Foster City, CA, USA) were employed for the sequencing reaction. Relevant bacterial DNA sequences of *Vibrio* spp. were obtained using the Applied Biosystems 3130 × Genetic Analyzer (HITACHI, Minato-ku, Tokyo, Japan). These sequences were analyzed for homology using the BLASTn program (http://www.ncbi.nlm.nih.gov/Blast (accessed on 3 January 2023)). Multiple alignment through Muscle tool and maximum likelihood phylogenetic analysis were conducted using the Mega software version 7.0 [39].

### 2.3. Susceptibility of Pathogenic Vibrio Strains to Antibiotics

The susceptibility of the two identified *Vibrio* strains (*V. harvei* and *V. alginoticus*) to antibiotics was screened by the in vitro agar diffusion method [40–42] using seven different antimicrobial disks (diameter 6 mm, Bioanalyse®, Ankara, Turkey) [chloramphenicol (C, 30 μg), ciprofloxacin (CIP, 5 μg), amoxicillin (AML, 25 μg), norfloxacin (NOR, 10 μg), doxycycline (DO, 30 μg), erythromycin (E, 15 μg), and florfenicol (F, 30 μg)], through an antibiogram. Purified colonies from each isolate were inoculated onto Mueller-Hinton agar (MHA) plates (Oxoid, Hampshire, UK) supplemented with 1.5% (*w/v*) sodium chloride using sterile cotton buds. Plates were incubated agar down between 24 and 48 h at 30 °C. The diameters of the inhibition halos surrounding the disks were measured in millimeters and interpreted as sensitive, intermediate, or resistant, according to the instructions of the Clinical and Laboratory Standards Institute [43] and the standardization of monitoring of antimicrobial usage and subsequent antimicrobial resistance in shrimp farming [17].

### 2.4. Experimental Control of Vibrio Infection by Antibiotic Application

After the antibiotic sensitivity test, the efficacy of in vivo antibiotic application against vibriosis in Pacific white shrimp was evaluated. A total of 120 alive, healthy, and infected broodstock shrimp (20 shrimp/group), which were divided into two groups (control and treated) in ss, were used in the experimental trial; each group was kept in a 20 L glass aquaria and supplied with hatchery water with mechanical aeration. For this purpose, florfenicol (10% Pan Flor, Marcyrl Animal Health, Cairo, Egypt) was mixed with a commercial feed (Skretting, Nutreco Company, Amersfoort, Netherlands) for diseased broodstock shrimp at a dose of 5 mg kg$^{-1}$ body weight. The antimicrobial was added to the commercial feed and vigorously homogenized, followed by the addition of 10 μL of soybean oil per gram of feed to avoid a lack of antimicrobial hydro-solubilization during feeding. The control group of healthy shrimp were fed a basal commercial feed. The diseased broodstock were fed the medicated diet at a rate of 5% of their body weight per day for 10 consecutive days. The cumulative mortality was recorded during the antibiotic application. Thereafter, the hemolymph and hepatopancreas (n = 6) from each group were sampled on day 11 as previously described for the immunological assessment and gene expression analysis, respectively.

### 2.5. Total Hemocyte Count

One hundred microliters of hemolymph-anticoagulant mixtures were incubated with 100 μL fixative solution (10% formalin in 0.45 M NaCl) for 20 min at room temperature. The hemocytes were counted using a light microscope and hemocytometer (Boeco, Hamburg, Germany) [44]. Fixed hemocyte suspensions were used for the preparation of smears. The smears were completely air-dried before staining with Giemsa solution. The numbers and



relative proportions of different hemocytes were calculated by counting at least 200 cells on each slide. The absolute differential hemocyte count was determined using the following equation: [(number of different hemocyte cell types/200) × THC] [45].

### 2.6. Enzyme Activity Assays and Antioxidant Index
2.6.1. Hepatopancreatic Enzymes

The activities of alanine and aspartate aminotransferases (ALT & AST) were tested in the supernatant of hemolymph using commercial kits (BioMed, Bayern, Germany) with a spectrophotometer (5010, Photometer, ROBERT RIELE GmbH & Co KG, Berlin, Germany). The optical density was evaluated at 546 nm.

2.6.2. Alkaline Phosphatase and Acid Phosphatase Enzymes and Antioxidants

Alkaline (ALP) and acid phosphatase (ACP) activities were evaluated using p-nitrophenyl phosphate (PNPP) as the standard substrate [46]. ALP and ACP were –estimated using p-nitro phenyl phosphate 16 mM as a standard substrate. Briefly, the substrate-hemolymph mixtures were incubated at 37 °C for 30 min after adding glycine NaOH and sodium acetate buffers for ALP and ACP, respectively. The released p-nitrophenol in the resulting supernatants was measured at 410 nm (5010, Photometer, ROBERT RIELE GmbH & Co KG, Berlin, Germany), and the amount was calculated from the standard curve.

The lysozyme activity was assessed following the previously described protocol [47] based on its ability to lyse *Micrococcus lysodeikticus* (Sigma Chemical Co., Saint Louis, MO, USA). Briefly, after centrifugation of the hemolymph-anticoagulant mixture, the precipitate was mixed with 0.02% *Micrococcus lysodeikticus* diluted in shrimp saline. The reaction was applied at room temperature (25 °C) and the absorbance was measured at 1 min intervals for 5 min at 540 nm using a microplate reader (Optica, Mikura Ltd., Horsham, UK). The lysozyme concentration was calculated using the calibration curve performed using lyophilized chicken egg-white lysozyme (Sigma Chemical Co., Saint Louis, MO, USA).

Superoxide dismutase (SOD) activity was estimated in the supernatant of hemolymph using the Ransod kit (Randox, Crumlin, UK) and evaluation of SOD's ability to inhibit superoxide radical-dependent reactions [48]. The reaction mixture consisted of xanthine and 2-(4-iodophenyl)-3-(4-nitrophenol)-5-phenyltetrazolium chloride (INT); the xanthine oxidase enzyme was prepared. The superoxide radicals resulting from the xanthine were directly reacted with INT to produce a red formazan dye. The optical density was evaluated at 505 nm (5010, Photometer, ROBERT RIELE GmbH & Co KG, Berlin, Germany). One unit of SOD was the amount desired to suppress the rate of xanthine reduction by 50%.

Glutathione (GSH) level was evaluated depending on its capacity to reduce 5,5′dithiobis or 5,5′-dithiobis (2-nitrobenzoic acid) (Sigma Chemical Co., Saint Louis, MO, USA) with GSH to produce a yellow compound [49]. This reduced chromogen is directly proportional to GSH concentration and its absorbance can be measured at 405 nm (5010, Photometer, ROBERT RIELE GmbH & Co KG, Berlin, Germany).

### 2.7. Non-Specific Immune Response Assays
2.7.1. Phagocytosis Percent Assay

The phagocytic activity was conducted following the protocol described in a previous study [50,51], with some modifications. Briefly, 500 μL of cell suspension ($5 \times 10^5$ cells mL$^{-1}$) were gently mixed with the same volume of latex beads (~$10^7$ beads mL$^1$; 1.094 μL; Polysciences Inc., Warrington, PA, USA) and incubated at room temperature for 30 min. Strictly, 200 μL of the hemocyte-bead mixture was mixed with 0.1 mL of 10% formalin for 20 min at room temperature. Twenty microliters of fixed-hemocyte were smeared on glass slides, air-dried, and fixed with methanol for 5 min before staining with Giemsa stain. Numbers of phagocytizing cells were counted from any 200 cells observed using a light microscope. The phagocytic activity was defined as follows: phagocytosis (%) = (number of cells ingesting beads/200) × 100.

2.7.2. Respiratory Burst Assay

The superoxide anion of hemocytes was estimated by measuring the formazan formed from the reduction of nitroblue tetrazolium (NBT) [52]. Briefly, hemolymph-anticoagulant mixtures (100 mL) collected from control, infected, and treated groups were pipetted into the wells of a microtiter plate and then incubated for 2 h at room temperature. After incubation, the supernatant was removed; the hemocytes were washed three times with Hank's buffered salt solution, and then one hundred microliters of NBT-PMA (NBT; 0.3% in PBS, 100 mL: phorbol 12-myristate 13-acetate PMA, Sigma-Aldrich; 1 mg mL$^{-1}$ PBS) were added to different wells and incubated for 30 min at 25°C. Later, the NBT solution was removed and absolute methanol was added. Then, the hemocytes were washed three times with 70% methanol and air-dried. Formazan deposits were dissolved by adding dimethyl sulfoxide (DMSO, Sigma) and 2 M potassium hydroxide (KOH, Sigma Chemical Co., Saint Louis, MO, USA). The respiratory burst activity was expressed as optical density (OD) that was measured at 630 nm using a microplate reader (Optica, Mikura Ltd., Horsham, UK).

2.7.3. Bactericidal Activity

The bactericidal activity of hemolymph supernatant (serum) was determined as described previously with some modifications [50]. A volume of the serum samples (100 μL) from each group were pipetted into different sets of wells with the same volume of *V. harveyi* bacterial suspension ($1 \times 10^8$ CFU mL$^{-1}$). Another set of wells containing shrimp PBS (sodium chloride 136 mM, potassium chloride 2.68 mM, disodium phosphate 8 mM, potassium dihydrogenphosphate 1.76 mM, in distilled water, pH 7.5) plus bacteria was prepared as a blank control. The microtiter plate was then incubated for 3 h at 25 °C before adding diphenyltetrazolium bromide solution (3 mg mL$^{-1}$ PBS) for a continuous 30 min incubation at room temperature. Later, the supernatant was completely removed, and the formazan precipitate was suspended in dimethyl sulfoxide (DMSO, Sigma Chemical Co., Saint Louis, MO, USA). The bactericidal activity was expressed as absorbance units (O.D.) after measuring the absorbance at 560 nm using a microtiter plate reader (Optica, Mikura Ltd., Horsham, UK).

*2.8. Total RNA Extraction, cDNA Synthesis, and Real-Time Quantitative PCR (qRT-PCR) Assay*

Total RNA from the tested hepatopancreas samples collected in RNA*later*® (Sigma-Aldrich, Saint Louis, MO, USA) from control, infected, and treated shrimp was extracted using Easy-Spin™ [DNA Free] Total RNA Extraction Kits (iNtRON Biotechnology, Inc., Sagimakgol-ro, Republic of Korea) according to the manufacturer's instructions. RNA integrity was confirmed by agarose gel electrophoresis, and the concentrations and purities of the samples were examined using a NanoDrop spectrophotometer (Thermo Fisher Scientific, Waltham, MA, USA). First-strand cDNA was synthesized in a reaction volume (20 μL) containing 1 μg of total RNA using a TOPscript™ cDNA Synthesis Kit (Enzynomics Co. Ltd., Sagimakgol-ro, Republic of Korea). Subsequently, *L. vannamei* hepatopancreas cDNAs (1 μL) were quantified using TOPreal™ 2× PreMIX SYBR Green qPCR master mix with low ROX (Enzynomics Co. Ltd., Daejeon, Republic of Korea) according to the manufacturer's recommendations using a Rotor-Gene Q MDx 6 plex real-time PCR system (Qiagen, Germantown, MD, USA). Specific primers, proPO (*LvproPO*), hemocyanin (*LvHc*), and toll-like receptor 1 (*LvToll1*) were employed to amplify the selected genes together with *β*-actin as the housekeeping gene (Table 1). The PCR program consisted of 40 cycles at 95 °C (10 s), 60 °C (15 s), and 72 °C (30 s), followed by melt curve generation at 72–95 °C for seconds. The relative gene expression levels were evaluated in triplicates included on template controls using the $2^{-\Delta\Delta CT}$ formula [53].

**Table 1.** Sequences of primer pairs used in the quantitative real-time PCR reactions.

| Gene | qPCR Primers (5′–3′) | Reference |
|---|---|---|
| β-Actin-F | CCACGAGACCACCTACAAC | [54] |
| β-Actin-R | AGCGAGGGCAGTGATTTC | |
| Lv PPO-F | CTGGGCCCGGGAACTCAAG | [55] |
| Lv PPO-R | GGTGAGCATGAAGAAGAGCTGGA | |
| Lv Hc-F | GTCTTAGTGGTTCTTGGGCTTGTC | [56] |
| Lv Hc-R | GGTCTCCGTCCTGAATGTCTCC | |
| LvToll1-F | TCGACCATCCCTTTTACACC | [57] |
| LvToll1-R | TTGCCTGGAAGGTCTGATTC | |

*2.9. Data Analysis*

Data related to immune parameters as well as gene expression were analyzed using GraphPad Prism v. 8.0 (GraphPad Software, Inc., San Diego, CA, USA) and statistically presented as mean ± standard error (SEM). The comparison of the data related to enzyme activities, the antioxidant index, and immunological parameters was performed using a one-way analysis of variance (ANOVA) followed by a Tukey's test among the three groups. Variations of the related gene expression values were estimated and applied. Fold changes were displayed using a log-base-two scale to highlight the changes in transcriptional levels, where $p < 0.05$ (*) and $p < 0.01$ (**) were considered to indicate significant differences, whereas ns was considered non-significant.

**3. Results**

*3.1. Clinical Signs of Diseased Shrimp*

Compared to healthy shrimp, the hepatopancreas and body of diseased *L. vannamei* were significantly reddish looking and lethargic, and the hepatopancreas exhibited a petechial hemorrhage in the abdominal muscle and red coloration of the body and pleopods (Figure 2A). Moreover, reddish spots appeared on the anterior carapace, including pleopods (Figure 2B).

*3.2. Characterization of Bacterial Vibrio Strains*

A total of 45 bacterial isolates of *Vibrio* spp. Were recovered from diseased *L. vannamei* based on bacterial biochemical and morphological characterization. The API 20E identification system sorted the isolates into *V. atlanticus* (5 isolates), *V. natriegens* (7 isolates), *V. alginolyticus* (15 isolates), and *V. harveyi* (18 isolates).

A PCR assay of bacterial isolates using *Vibrio* 16S rRNA primers showed amplified bands of a 663-bp size that was characteristic of *Vibrio* spp. Isolated from infected *L. vannamei* (Figure 3). The amplicon sizes of the band obtained from the positive control sample corresponded to the predicted sizes, whereas no amplicon was observed in the negative control (Figure 3).

In this study, the BLAST analysis of the nucleotide sequence of the four 16S rRNA genes from *Vibrio* spp. Shared 97.31–98.5% identity with the 16S rRNA partial sequences and the complete genomes of *V. atlanticus*, *V. natriegens*, *V. alginolyticus*, and *V. harveyi*, respectively, obtained from the GenBank database. Based on the phylogenetic tree constructed from 16S rRNA gene partial sequences, *Vibrio* strain-1 was grouped with *V. atlanticus* isolates; however, *Vibrio* strain-2 was grouped with a different *Vibrio* spp., including *V. natriegens* (Figure 4). Meanwhile, *Vibrio* strain-3 and strain-4 were grouped with *V. alginolyticus* and also *V. harveyi* (Figure 4), respectively. No bacterial load was detected on the samples isolated from the treated and control shrimp.

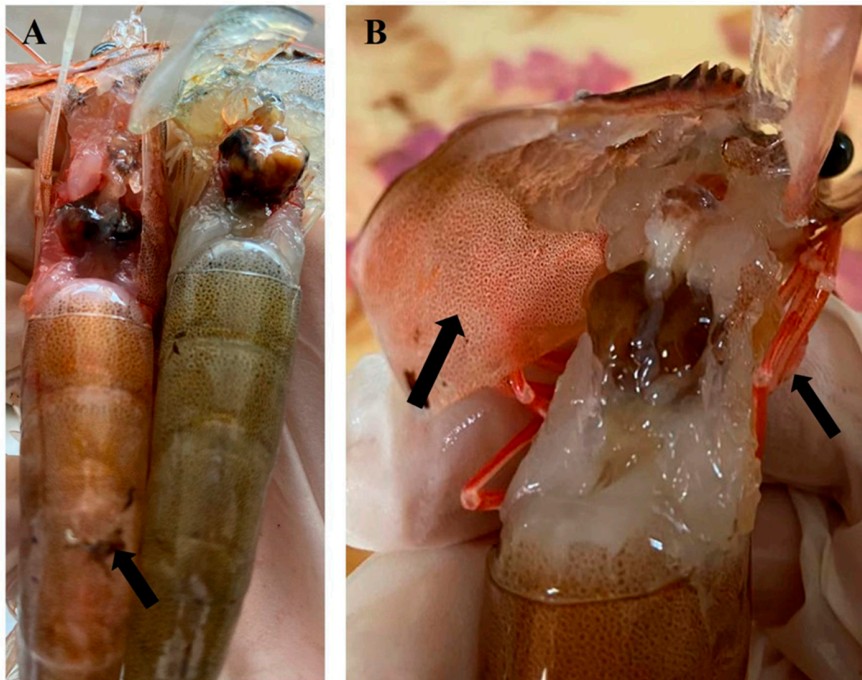

**Figure 2.** (**A**): Hatchery field cases of infected and normal broodstock shrimp, *Litopenaeus vannamei*. (**B**): infected shrimp show reddish discoloration of abdominal muscles, hepatopancreas, and pleopods, with a melanized area on the reddish abdominal cuticle (arrow). Red-colored spots are shown on the carapace with pereopods (arrow).

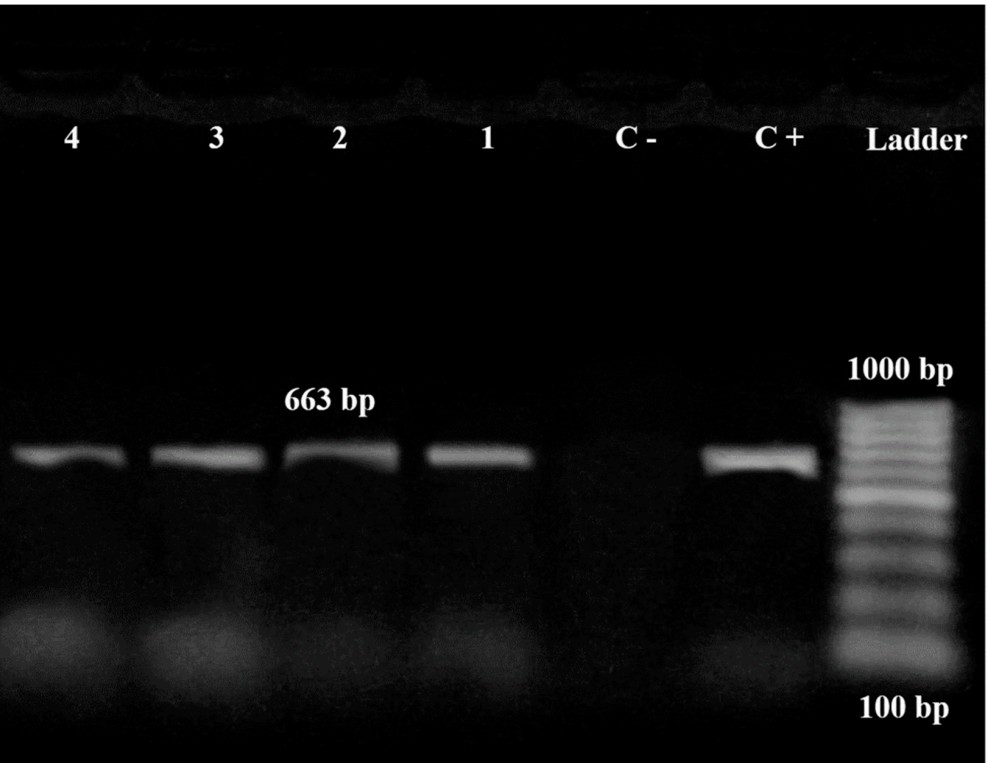

**Figure 3.** PCR amplification of the 16S rRNA gene of *Vibrio* spp. Isolated from *Litopenaeus vannamei*. The PCR products displayed corresponded to the predicted molecular mass of 663-bp (16S rRNA gene). Lane (C+): positive control sample. Lane (C−): negative control, and to its right is the 100-bp DNA ladder. Lanes (1–4): represent the bacterial DNA samples.

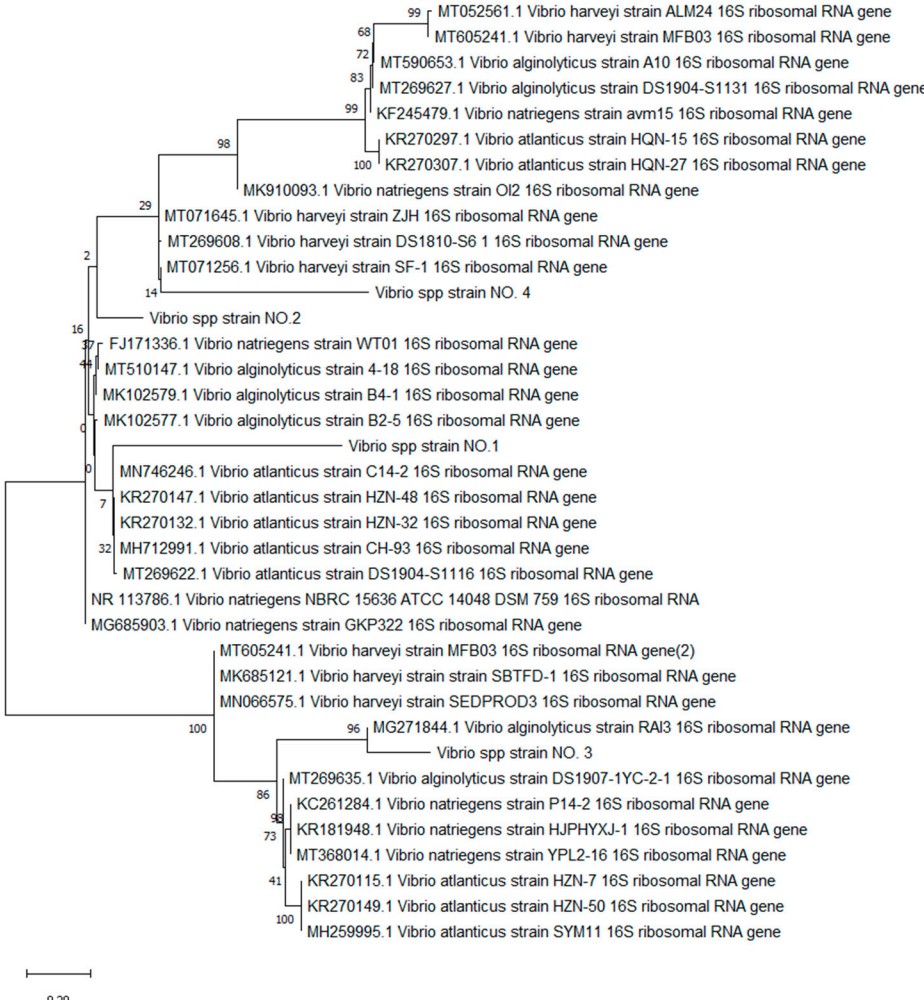

**Figure 4.** Phylogenetic tree of the four *Vibrio* strains isolated from diseased shrimp, *Litopenaeus vannamei* was constructed using maximum likelihood based on the 16S rRNA sequences of *Vibrio* spp. The numbers above the branches are the values calculated through a bootstrap analysis (1000 replicates).

### 3.3. Susceptibility of Pathogenic Vibrio Strains to Antibiotics

Bacterial *Vibrio* strains were highly sensitive to chloramphenicol, doxycycline, and florfenicol but resistant to amoxicillin. They exhibited moderate sensitivity to norfloxacin. The other *Vibrio* strain was moderately sensitive to erythromycin and ciprofloxacin; however, the other strain showed dissimilar results (Table 2).

Cumulative mortality rates were initially estimated to be around 55% on day 1, which continued at the same percent for the successive five days. Subsequently, the FLO application decreased the cumulative mortality to 20% for the medicated shrimp. There were no mortalities in the non-infected broodstock shrimp (control group).

### 3.4. Total Hemocyte Count

Total hemocyte count (THC), granular cell (GC), semi-granular cell (SGC), and hyaline cell (HC) counts of all experimental groups are shown in Figure 5. The *Vibrio*-infected group exhibited a significant drop in the THC ($t = 10.58$; $df = 8$; $p = 0.0001$), GC ($t = 9.12$; $df = 8$; $p = 0.0001$), SGC ($t = 10.55$; $df = 8$; $p = 0.0001$), and HC ($t = 9.73$; $df = 8$; $p = 0.0001$) count compared to the control group ($p < 0.001$). However, the treatment of infected shrimp with florfenicol significantly restored the total hemocyte cell counts ($t = 5.86$; $df = 8$; $p = 0.0004$), GC ($t = 7.32$; $df = 8$; $p < 0.0001$), SGC ($t = 7.11$; $df = 8$; $p = 0.0001$), and HC ($t = 7.19$; $df = 8$; $p < 0.0001$) counts to their normal levels, relative to the infected group (Figure 5).

**Table 2.** Susceptibility of two pathogenic *Vibrio* strains to antibiotics through the results of the antibiogram tests.

| Antibiotic | Concentration (µg) | *Vibrio harveyi* | *Vibrio alginolyticus* |
|---|---|---|---|
| amoxicillin (AML) | 25 | R | R |
| erythromycin I | 15 | s | R |
| chloramphenicI(C) | 30 | S | S |
| doxycycline (DO) | 30 | S | S |
| florfenicol (FFC) | 30 | S | S |
| ciprofloxacin (CIP) | 5 | s | S |
| norfloxacin (NOR) | 10 | s | s |

The sensitivity criterion used was based on the standardization of surveillance of antimicrobial usage and antimicrobial resistance in shrimp aquaculture. S: Highly sensitive; R: Resistant; s: moderately sensitive.

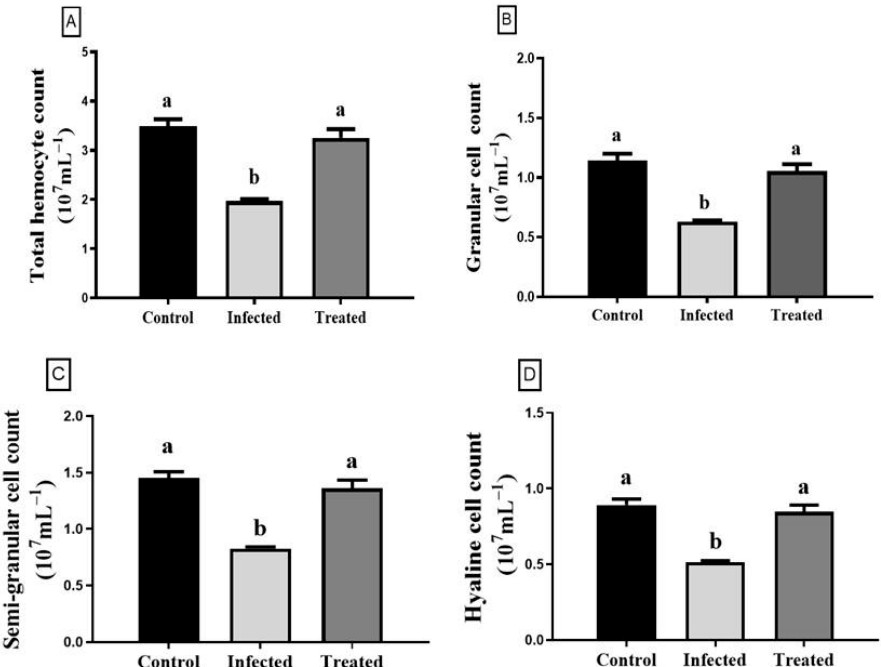

**Figure 5.** Total hemocyte count (THC; (**A**)), granular cell (GC; (**B**)), semi-granular cell (SGC; (**C**)), and hyaline cell (HC; (**D**)) of infected *Litopenaeus vannamei* with *Vibrio* spp. and those treated with florfenicol antibiotic. Values are expressed as the mean ± SME (n = 6 per group). Means with different superscripts are significantly different ($p < 0.05$).

*3.5. Enzyme Activity Assays and Antioxidant Index*

The enzyme activities and antioxidants estimated in the supernatant of hemolymph are enumerated in Figures 6 and 7. The ALT ($t = 8.31$; $df = 8$; $p = 0.0001$), AST ($t = 8.52$; $df = 8$; $p = 0.0001$), ALP ($t = 7.19$; $df = 8$; $p < 0.0001$), and ACP ($t = 18.68$; $df = 8$; $p < 0.0001$) activities were significantly elevated in *L. vannamei* infected with *Vibrio* spp. compared to the control (Figure 6). Meanwhile, the previous enzyme activities (ALT ($t = 2.25$; $df = 8$; $p = 0.0542$), AST ($t = 6.81$; $df = 8$; $p = 0.0001$), ALP ($t = 6.67$; $df = 8$; $p = 0.0002$), and ACP ($t = 14.58$; $df = 8$; $p < 0.0001$) were significantly reestablished in the antibiotic-treated group compared to the infected group (Figure 6).

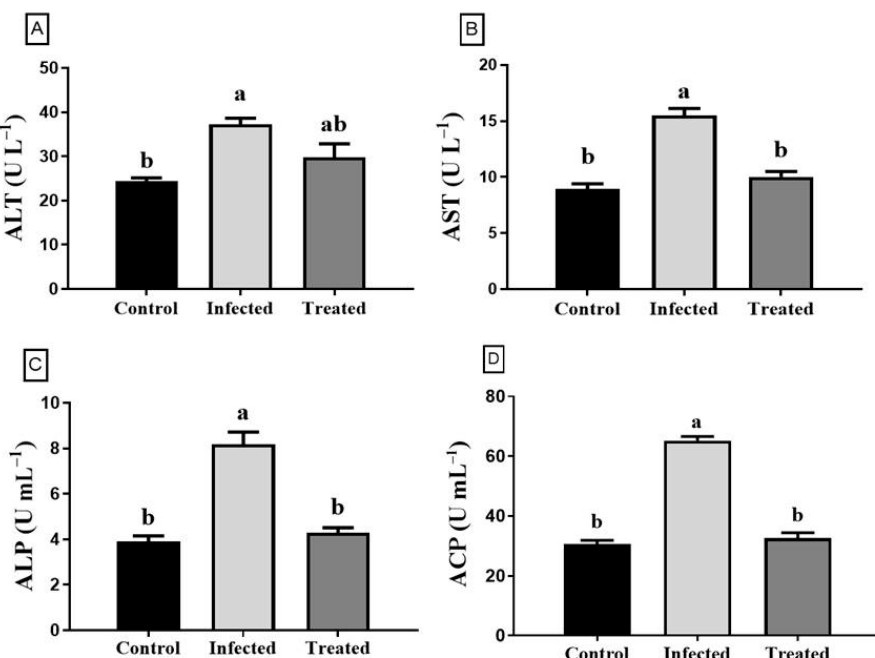

**Figure 6.** Alanine aminotransferase (ALT; (**A**)), aspartate aminotransferase (AST; (**B**)), alkaline phosphatase (ALP; (**C**)), and acid phosphatase (ACP; (**D**)) activities of infected *Litopenaeus vannamei* with *Vibrio* spp. and those treated with florfenicol antibiotic. Values are expressed as the mean $\pm$ SME (n = 6 per group). Means with different superscripts are significantly different ($p < 0.05$). Lysozyme activity ($t = 10.17$; $df = 8$; $p < 0.0001$), SOD ($t = 11.91$; $df = 8$; $p < 0.0001$), and GSH ($t = 6.14$; $df = 8$; $p = 0.0003$) indexes were significantly lower in *Vibrio*-infected shrimp ($p < 0.001$) than in the control group (Figure 7). The treatment with florfenicol elevated the lowered levels of lysozyme ($t = 6.20$; $df = 8$; $p = 0.0003$), SOD ($t = 8.76$; $df = 8$; $p < 0.0001$), and GSH ($t = 5.58$; $df = 8$; $p = 0.0005$) caused by *Vibrio* infections as demonstrated in the treated group when compared to the infected group (Figure 7).

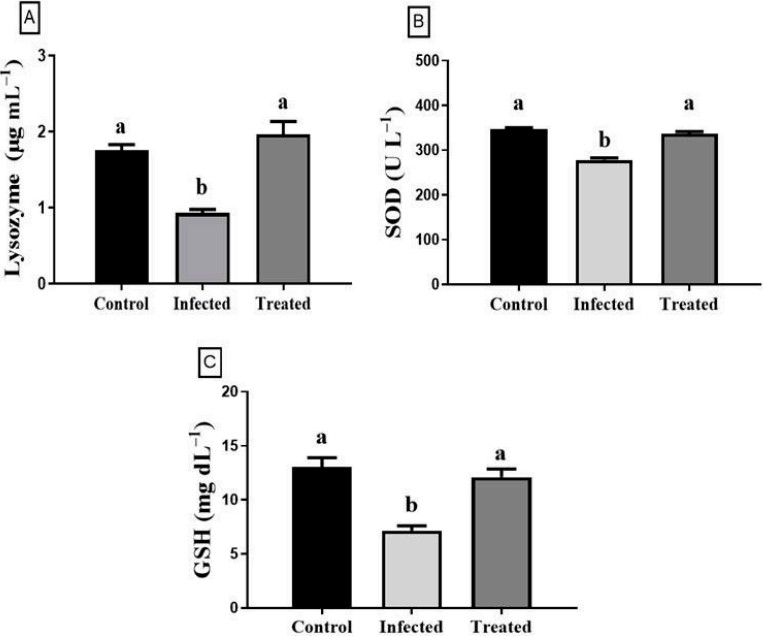

**Figure 7.** Lysozyme activity (**A**), superoxide dismutase (SOD; (**B**)), and glutathione (GSH; (**C**)) of infected *Litopenaeus vannamei* with *Vibrio* spp. and those treated with florfenicol antibiotic. Values are expressed as the mean $\pm$ SME (n = 6 per group). Means with different superscripts are significantly different ($p < 0.05$).

### 3.6. Immune Parameters

The effects of the florfenicol treatment on the phagocytic, respiratory burst, and bactericidal activities of shrimp against *Vibrio* spp. infection are displayed in Figure 8. Phagocytosis % ($t = 9.60$; $df = 8$; $p < 0.0001$), respiratory burst ($t = 6.18$; $df = 8$; $p = 0.0003$), and bactericidal ($t = 6.67$; $df = 8$; $p = 0.0002$) activities of shrimp treated with florfenicol were significantly increased compared to those of *Vibrio*-infected shrimp which represented significant suppression in the different immune responses (Figure 8).

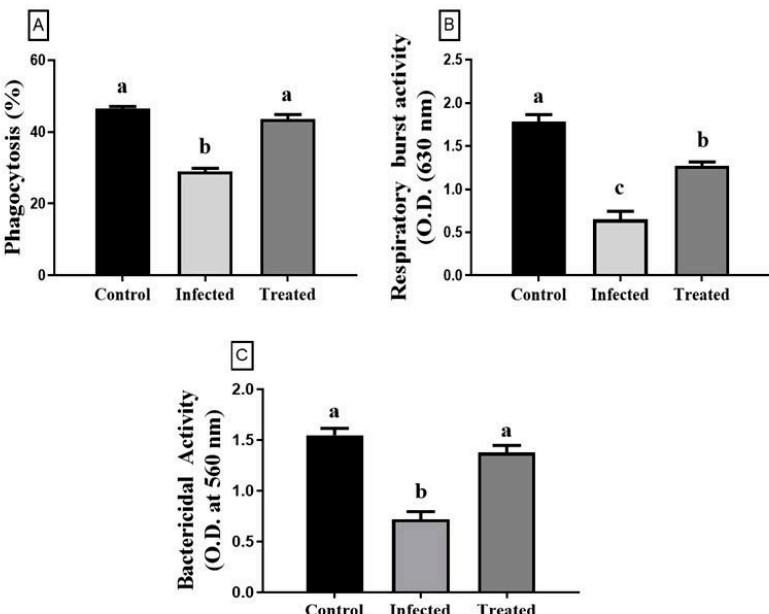

**Figure 8.** (**A**): Phagocytosis (%). (**B**): Respiratory bursts, and (**C**): Bactericidal activities of infected *Litopenaeus vannamei* with *Vibrio* spp. and those treated with florfenicol antibiotic. Values are expressed as the mean $\pm$ SME (n = 6 per group). Means with different superscripts are significantly different ($p < 0.05$).

### 3.7. Expression Profiles of LvHc, LvToll1, and LvproPO

The transcript levels of the *LvproPO* and *LvToll1* genes in the hepatopancreas were not statistically different between groups (Figure 9A,C) although lower expression levels were observed in the infected and treated groups compared with the control group. Similarly, the mRNA expression of *LvHc* in the hepatopancreas was markedly downregulated in the treated group ($p = 0.0103$, $0.0038$) when compared with the control and infected groups (Figure 9B).

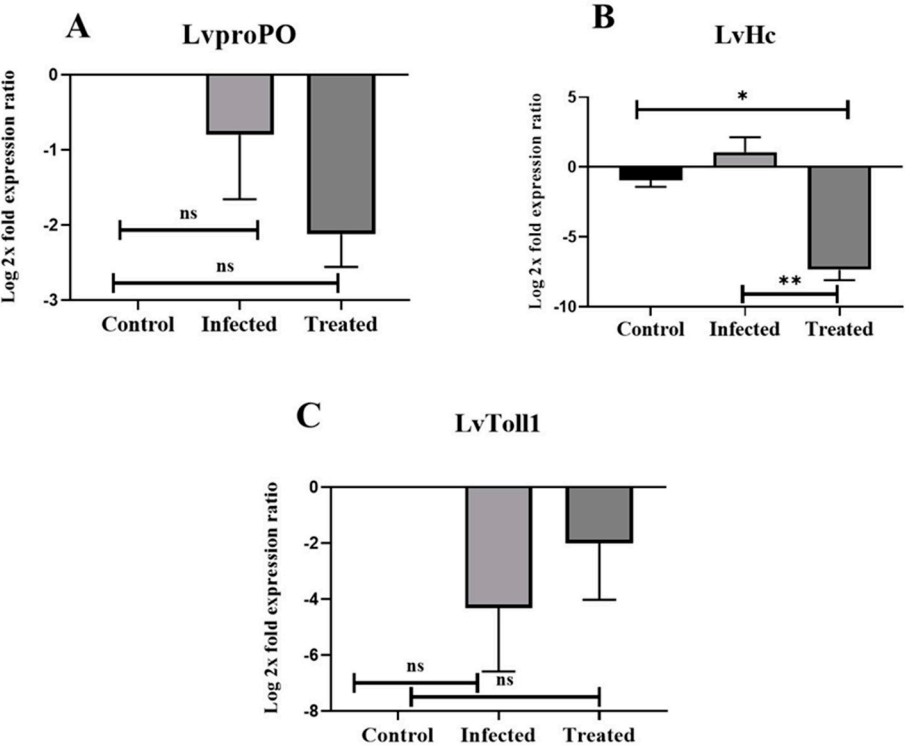

**Figure 9.** Relative qPCR expression analysis of the *LvproPO* (**A**), *LvHc* (**B**), and *LvToll1* (**C**) genes in the hepatopancreas of *Litopenaeus vannamei* in the control, infected, and treated groups. Data are normalized to *β*-actin. The data were analyzed by a one-way ANOVA and presented as a fold change between groups. Values are expressed as the mean ± SE (n = 3 per group). Asterisks refer to significant differences between groups at * $p < 0.05$, ** $p < 0.01$, and ns: non-significant.

## 4. Discussion

*Vibrio* species remain a life-threatening pathogen in shrimp hatcheries that have a crucial demand for shrimp culture. Owing to vibriosis, several countries, such as India, Thailand, and Mexico, have reported severe economic losses in *L. vannamei* aquaculture [58,59]. The stress associated with diminished resistance to *Vibrio* spp. infection has primarily promoted high mortalities in shrimp culture, especially diseases caused by *V. harveyi* and *V. alginolyticus* [12,60].

In the present study, *Vibrio* strains were identified by 16S rRNA gene sequencing from naturally infected *L. vannamei* broodstocks in a shrimp hatchery in the Egyptian coastal province of DTZ. Among these, *V. harveyi* has been previously reported to promote bright-red syndrome in Pacific white shrimp [58,59]. In addition, *V. natriegens* has been reported to cause high mortalities in shrimp culture [61]. Moreover, diseased shrimp displayed cuticular lesions with a red coloration of the body, hepatopancreas, and pleopods; these symptoms are similar to those of shrimp bright-red syndrome, especially the disease caused by *V. harveyi* and *V. alginolyticus* [9,56].

Regarding the phylogenetic analysis, most of the isolated strains retrieved from the diseased *L. vannamei* shared almost identical sequences in the 16S rRNA gene (98.8% identity) and belonged to the *Harveyi* clade, consisting of *V. harveyi*, *V. alginolyticus*, and *V. natriegens*, which shared high levels of phenotypic and genotypic homology and are known to be pathogenic for shrimp [13,62]. This finding was consistent with the results observed in the Ecuadorian *L. vannamei* hatcheries suffering mortalities [26]. However, new strains are identified in our studies, such as *V. atlanticus* and *V. natriegens* in Egypt, suggesting the diversity of the circulating bacteria and indicating the crucial importance of studying the effectiveness of the shrimp immune response.

Antibiotics are the most effective therapeutic agents against several pathogenic bacteria in shrimp farming [63]. We evaluated several antibiotics in our study to demonstrate the

patterns of antibiotic resistance; however, most of these antibiotics are not authorized for use in aquaculture. In our screening, both strains were resistant to amoxicillin similar to reports of several resistant *Vibrio* spp. isolated from diseased shrimp to first-generation penicillins or/and $\beta$-lactam antibiotics [59,64]. Moderate sensitivity was shown for enrofloxacin, which is one of the currently used antibiotics in several countries against vibriosis [58,59]. Interestingly, *Vibrio* strains have been documented to be sensitive to most of the tested antibiotics. Florfenicol is used in our study to decrease the mortality rates of shrimp owing to its potent activity against bacterial pathogens. In general, FLO was also approved by the FDA for use in shrimp aquaculture because of its high potency for the control of necrotizing hepatopancreatitis and vibriosis infections in farm-raised Pacific white shrimp [65]. Many studies have used FLO for controlling the mortality caused by *Vibrio* spp. isolated in Ecuador, the USA, Japan, Thailand, and Mexico [66,67]. Additionally, absorption of FLO from the intestine has been shown to be rapid with extensive distribution and prompt elimination in Pacific white shrimp, *Litopenaeus vannamei* [68]. A recent study showed that low concentrations of florfenicol (minimal inhibitory concentration, MIC of 8 $\mu$g mL$^{-1}$) were highly effective in controlling bacterial growth for many pathogenic *Vibrio* strains [26]. Moreover, FLO had a greater favorable effect on the survivability of adult shrimp (*Litopenaeus vannamei*) with necrotizing hepatopancreatitis (NHP) disease by diminishing the bacterial load, allowing the shrimp to mount an immune response to the pathogens [27]. However, it has been reported that oral administration of FLO at 100 and 200 mg kg$^-$ had a suppressive effect on the antioxidant activity of juvenile *Litopenaeus vannamei* [69]. Regarding our results from this study, it is reasonable to assume that a dose of 5 mg kg$^-$ given orally for FLO should be applicable for the control of vibriosis in Pacific white shrimp.

Hemocytes play a critical role in the immune response of crustaceans, including phagocytosis, mediation of cytotoxicity, encapsulation, and nodule formation [70]. The hemocytes are categorized into three types based on the presence of granules or relative size: large granular (granular, GC), small granular (semi-granular, SGC), and granular (hyaline, HC) hemocytes [32]. The decline in THC counts in infected *L. vannamei* was revealed in our study. Generally, exposure to infectious pathogens or environmental stress reduces THC counts in shrimp, which in turn boosts the risk of secondary infection [71]. On the other hand, the THC of shrimp declined in softshell clams, *Mya arenaria* infected with *V. splendidus* [72], and *L. vannamei* infected with *V. harveyi* [73]. This decline in THCs was associated with the bacterial inflammatory response as hemocytes leave the circulation and migrate to the site of infection [74]. As well, hemocytes aggregate into hemocytic nodules with cell adhesion molecules to eliminate bacteria from the circulation [34]. The current study showed that florfenicol medication could reestablish THC, GC, SGC, and HC in the white shrimp infected with *Vibrio* spp. A previous report enumerated that the supplementation of chitosan–gentamicin conjugates increased THCs in *L. vannamei* infected with *V. parahaemolyticus* through enhancement of hemocyte proliferation and their phagocytic activity [75]. Enrofloxacin promoted disease resistance against *V. parahaemolyticus* and restored THCs in shrimp [76].

The main biomarkers for hepatopancreas function, AST and ALT activities, can reflect the degree of hepatopancreas injury in shrimp [77]. The elevation of AST and ALT activities suggested possible damage to the hepatopancreatic tissue as in necrotizing hepatopancreatitis disease caused by *Vibrio* spp., which was significantly decreased following oxytetracycline administration in *L. vannamei* [27]. In our study, shrimp treated with florfenicol had significantly lower hepatopancreatic enzyme levels (ALT and AST) compared to the infected and control groups [78]. During infection with *V. parahaemolyticus*, the epithelial cells of the hepatopancreatic tubules rupture following infection. When treated with florfenicol, the hepatopancreas showed intact epithelial cells with the improved structural integrity of the tubules, explaining the decrease in the hepatopancreas enzyme activity in the hemolymph [78].

Various enzymes in the hemolymph, such as ALP, ACP, SOD, and lysozyme, are considered indicators for the evaluation of disease resistance and immune status in shrimp [79]. ALP is one of the regulatory enzymes connected to the metabolism process and phagolysis; whereas, ACP is an identical lysosomal enzyme that has a key role in the elimination and hydrolyzing of microbes [80,81]. Both ALP and ACP are involved in the regulation of the phosphorylation and dephosphorylation processes [82]. The ACP activity of *Chlamys farreri* was significantly elevated following the challenge with *V. anguillarum* [83]. Other comparable studies demonstrated that ACP activities of *L. vannamei* are more sensitive to *V. parahaemolyticus* and their increase is attributed to the disturbance of cell metabolism and immune function as well as the stress caused by *Vibrio* infection [75,76,84], which is consistent with our results. Moreover, the administration of florfenicol significantly lowered ALP and ACP activities, which may be due to its ability to strengthen resistance against infection as the activity of these immune enzymes was significantly elevated in the cell-free hemolymph of shrimp following *Vibrio* infection in *L. vannamei* [78].

Lysozyme is a functional antibacterial protein and a pivotal component of the invertebrates' innate immune system as it can hydrolyze the mucopolysaccharides present in the bacterial cell wall to kill pathogens [85]. SOD is one of the remarkable antioxidant enzymes that protect the host against oxidative stress through the degradation of excess $O^{2-}$ to produce molecular oxygen and hydrogen peroxide [86]. GSH is a valuable antioxidant in cells that can reduce hydrogen peroxide to water together with glutathione peroxidase to maintain the integrity of the red blood cell membrane and prevent damage by oxidants [87]. Our results exhibited lowered activities of lysozyme, SOD, and GSH following *Vibrio* infection, which were then restored following antibiotic administration. Similarly, previous reports presented a significant decrease in SOD activity in *L. vannamei* injected with *V. parahaemolyticus* and *V. alginolyticus*, respectively [84,88]. In contrast to our results, previous studies demonstrated increased lysozyme activity after *V. parahaemolyticus* and *V. alginolyticus* infection in *L. vannamei* and *P. trituberculatus* [76,89]. The lower GSH level and SOD activity may be correlated to the oxidative stress mediated by singlet oxygen that causes SOD inactivation during infection [90]. Whereas, lower lysozyme activity may be correlated with the inactivation of the immune response in shrimp challenged with *Vibrio* spp. [27]. Similar to our results, the SOD activity and GSH level of *L. vannamei* infected with *V. parahaemolyticus* were restored upon the oral administration of chitosan–gentamicin conjugate due to a great reduction in ROS production and lipid peroxidation after antibiotic treatment [75]. Dietary administration of a low dose of *Astragalus* and florfenicol increased lysozyme activity levels in shrimp challenged with *V. parahaemolyticus* [78].

Phagocytosis is the initial internal defense mechanism against any foreign objects, and one of the fundamental roles of hemolymph in the invertebrate defense process [91]. The phagocytic cells produce superoxide anion and its reactive derivatives that have powerful bactericidal activity, during the mechanism of respiratory burst [50]. Our results showed that infection with *Vibrio* reduced the phagocytosis and bactericidal activity of *L. vannamei*, as also demonstrated by previous studies [92,93]. The reduced phagocytosis of *V. splendidus*-challenged hemocytes could result from the loss of pseudopodia due to the toxic effect of extracellular products produced by bacteria [72]. *V. tapetis* also decreased the phagocytosis activity of hemocytes in the Japanese carpet clam *R. philippinarum* [94]. The lower respiratory burst activity may be due to either *V. splendidus* lacking a receptor that stimulates respiratory burst activity or it actively curbs the hemocyte's response [72]. In the current study, there was a significant decline in the phagocytic, respiratory burst, and bactericidal activities, which returned to normal in florfenicol-treated shrimp.

We detected slight differences in the transcript levels of the immune response genes in the hepatopancreas of *L. vannamei*. Activation of the *proPO* system during infection inhibits the damage caused by the proteases synthesized by pathogens [95]. Our study revealed that expression of the *proPO* gene displayed a decreasing pattern in the infected and treated groups. This result indicates that once pathogens are removed through the proPO system during infection or treatment, protease inhibitors capture the prophenoloxidase-activating

enzyme (proPOAE), and thus counteract the proPO-activation to the phenoloxidase isoform [96]. This result is consistent with other studies that reported the modulation of this gene in *L. vannamei* challenged with *V. harveyi* and the white spot syndrome virus [56,97]. ProPOs are primarily expressed in *L. vannamei* hemocytes but are present at very low levels in the hepatopancreas, as assessed by RT-PCR and Northern blotting analysis [54,97]. Likewise, the significant decrease in the expression of *LvHc* in our study can justify the lower expression levels of *LvproPO*, i.e., when hemocyanin proteolysis is generated, it is converted into a phenoloxidase-like enzyme, possessing a phenoloxidase activity [98]. The downregulation of immune-related genes, such as *penaeidin* and *proPO*, has been reported after oxytetracycline and oxolinic acid treatment in shrimp [99]. A similar trend regarding the expression of *LvToll1* was observed in our study, with no significant changes being detected in the treated group compared with the control group. However, shrimp tolls are involved in the regulation of AMPs as they are primarily synthesized in the hemocytes. Our results are consistent with another study that demonstrated that *LvToll1* could have other potential roles in the penaeid immune response, and the expression level of *LvToll1* was also low in the hepatopancreas of *L. vannamei* [57]. The results obtained in the FLO application sought greater effectiveness for the control of bacterial infections in shrimp hatcheries. However, we need to highlight the importance of withdrawal times for the antibiotic, with respect to eliminating the residual presence of these compounds from the edible tissues, and from the cultivation system to diminish the development of antibiotic resistance in the bacteria.

## 5. Conclusions

In summary, four *Vibrio* strains were isolated in this study from the hepatopancreas of *L. vannamei* shrimp in an Egyptian hatchery-raised shrimp. FLO supplementation scavenged most of the hemato-immune parameters as well as the pathogen load in *L. vannamei* shrimp infected with *Vibrio* spp. Herein, our data might evaluate the immunological screening of hatchery-diseased shrimp in indicating the effect of antibiotic medication in the management and control of vibriosis through regaining the immune response of *L. vannamei* shrimp after the antibiotic clears the infection rather than a direct response. Further investigation and exploration should be undertaken on the quality and usage of antibiotics in shrimp aquaculture.

**Author Contributions:** S.E. and G.E.E.: Methodology, Formal analysis, Investigation, Validation, and Writing—original draft, Writing—review and editing. S.E.: Conceptualization, Supervision, and Final revision. M.S.S.: Investigation, Visualization, and Methodology. A.A.A., E.M.Y.: Funding acquisition, Resources, Validation, Writing—review & editing. S.J.D.: Conceptualization, data support & analysis, Scientific critical input & Citation linkage, Literature comparative dialogue, Revisions. Funding acquisition and Resources. All authors have read and agreed to the published version of the manuscript.

**Funding:** This research was funded by the researchers supporting project number (RSPD2023R700), King Saud University, Riyadh, Saudi Arabia.

**Institutional Review Board Statement:** This study abides by the Medical Research Ethics Committee of Mansoura University and follows the general guidelines of the Canadian Council on Animal Care with the code number (R/65).

**Informed Consent Statement:** Not applicable.

**Data Availability Statement:** The data that support the findings of this study are available on reasonable request from the corresponding author, S. Elbahnaswy. The data are not publicly available due to their containing information that could compromise the privacy of research participants.

**Acknowledgments:** This research was funded by the researchers supporting project number (RSPD2023R700), King Saud University, Riyadh, Saudi Arabia.

**Conflicts of Interest:** The authors do not report any financial or personal connections with other persons or organizations that might negatively affect the contents of this publication and/or claim authorship rights to this publication.

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
