# Peer review of "Comparison of Immune Response of Litopenaeus vannamei Shrimp Naturally Infected with Vibrio Species, and after Being Fed with Florfenicol"

_fishes, doi:10.3390/fishes8030148_

Round 1
Reviewer 1 Report
General comments:
- Extensive editing of English language and style required.
- Authors need to make sure to add the city, state, and country of the manufacturer for all feed, material, chemicals, and equipment they used. Such as lines 116, 143, 149, 156, ...etc
- Make sure that L. vannamei is italic throughout the manuscript such as line 124
- Avoid using parentheses with parentheses such as in lines 20 and 22
- Avoid using parentheses back to back such as in line 392, 402, 404, 412, 415
Specific comments:
Abstract
- Line 14: four instead of 4
- line 20: these abbreviations were not spelled ou before this point "THC, GC, SGC, and HC)"
- lines 20 and 22: Avoid using parentheses with parentheses
Lines 32 to 34: In the abstract, the research objective comes before the results.
Introduction:
- Line 48: Are these annual losses? Are these the estimated losses in Egypt? Please add text to make it clear
- Lines 63-64: this sentence is not relevant to this study
_ Line 103: florfenicol instead of Florfenicol
Materials and Methods
- Line 109: add m for each dimension
- Line 123: Early in this paragraph, authors mentioned that diseases broodstock were collected which reflects that these were naturally infected. Here, they mentioned "within two days of infection"...what infection. It is confusing? how did the authors determine the time of infection?
- Line 136 (figure 1):
Line 137: add "by" after surrounded.
Line 141: add "shrimp" after "diseased"
Line 144: remove the lines under the Celsius symbol
Data analysis:
- It is not clear why immune parameters were analyzed using SPSS while gene expression values were analyzed using GraphPad?
- Authors used t-test to compare control vs infected, then another t-test to compare infected vs treated. Authors need to reanalyze data using ANOVA instead of using two t-tests
-Spell out the abbreviation "ANOVA" where it was mentioned for first time
- Line 327: "p < 0.01" was repeated twice
Results:
- For all statistical analysis results, authors need to report test statistics values (t or F), degrees of freedom, and the exact p values not just <0.05 or < 0.01.
-
Author Response
Comments and Suggestions for Authors
Manuscript: Fishes-2174464
Report 1
General comments:
- Extensive editing of English language and style required.
Manuscript has been checked by native English reviewer in details. All spelling and grammar mistakes have been corrected. In addition, we send the manuscripts to Egyptian Knowledge Bank (EKB) for English editing. We attached the certificate from EKB editing team.
- Authors need to make sure to add the city, state, and country of the manufacturer for all feed, material, chemicals, and equipment they used. Such as lines 116, 143, 149, 156, ...etc
We have added most of the information, however we could not find city or state of some reagents.
- Make sure that L. vannamei is italic throughout the manuscript such as line 124
We have modulated it according to your kind suggestions (Highlighted).
- Avoid using parentheses with parentheses such as in lines 20 and 22
We have modulated it.
- Avoid using parentheses back to back such as in line 392, 402, 404, 412, 415
We have modulated these parts.
Specific comments:
Abstract
- Line 14: four instead of 4
We have modulated it.
- line 20: these abbreviations were not spelled out before this point "THC, GC, SGC, and HC)"
We have modulated these parts.
- lines 20 and 22: Avoid using parentheses with parentheses
We have modulated it.
Lines 32 to 34: In the abstract, the research objective comes before the results.
We modulated this part.
Introduction:
- Line 48: Are these annual losses? Are these the estimated losses in Egypt? Please add text to make it clear
We have clarified this part.
- Lines 63-64: this sentence is not relevant to this study
We removed this sentence.
_ Line 103: florfenicol instead of Florfenicol
We have modulated this part
Materials and Methods
- Line 109: add m for each dimension
We added m according to your suggestion.
- Line 123: Early in this paragraph, authors mentioned that diseases broodstock were collected which reflects that these were naturally infected. Here, they mentioned "within two days of infection"...what infection. It is confusing? how did the authors determine the time of infection?
We have determined the initial infection from the owner of the shrimp hatchery from the exact day of the mortality of the shrimp during the outbreaks.
- Line 136 (figure 1):
Line 137: add "by" after surrounded.
Line 141: add "shrimp" after "diseased"
Line 144: remove the lines under the Celsius symbol
We have modulated these parts
Data analysis:
- It is not clear why immune parameters were analyzed using SPSS while gene expression values were analyzed using GraphPad?
We have cleared these parts
- Authors used t-test to compare control vs infected, then another t-test to compare infected vs treated. Authors need to reanalyze data using ANOVA instead of using two t-tests
We re-analysed the data using ANOVA test.
-Spell out the abbreviation "ANOVA" where it was mentioned for first time
We added the missing word for ANOVA
- Line 327: "p < 0.01" was repeated twice
We added the missing point
Results:
- For all statistical analysis results, authors need to report test statistics values (t or F), degrees of freedom, and the exact p values not just <0.05 or < 0.01.
We have added the required data.

Reviewer 2 Report
1. This study tested two pathogenic Vibrio strains for the susceptibility test excluded virulent V. natriegens that has been isolated, unfortunately. Those two pathogenic Vibrio strains used in this test, are they belong to isolates you identified in this study? It should be clearly mentioned. What about the non-Vibrio, normal shrimp microflora are most of them sensitive or resistant to florfenicol?
2. Have you done the metagenomic analysis of the microbial composition alteration and comparison beetween those three treatments in the study? Can be done with NGS, DGGE analyses, for example). Since many studies shown us that microbial composition is essential in the non specific immune system of the aquatic organism.
3. Vibrio strain-2 was grouped with different Vibrio spp. including V. natriegens (line 363), Where exactly is the position of isolate NO 2 in the Phylogenetic Tree?
4. The treatment of infected shrimp with florfenicol significantly restored the total and differential hemocyte cell counts to their normal levels, relative to the infected group (line 390-391). What about the role or contribution of the florfenicol-resistant beneficial microbes in shrimp immune system stimulation?
Author Response
Comments and Suggestions for Authors
- This study tested two pathogenic Vibrio strains for the susceptibility test excluded virulent V. natriegens that has been isolated, unfortunately. Those two pathogenic Vibrio strains used in this test, are they belong to isolates you identified in this study? It should be clearly mentioned. What about the non-Vibrio, normal shrimp microflora are most of them sensitive or resistant to florfenicol?
According to your kind suggestion, we can perform the antibiotic sensitivity test on the other isolates of Vibrio in a further study as we focused only on the most pathogenic stains, which constituted most of isolates.
Moreover, we should carry out a speedy and fast an antibiotic sensitivity test for the control of shrimp mortalities in the farm.
- Have you done the metagenomic analysis of the microbial composition alteration and comparison beetween those three treatments in the study? Can be done with NGS, DGGE analyses, for example). Since many studies shown us that microbial composition is essential in the non specific immune system of the aquatic organism.
We will perform the next generation sequencing of the isolated Vibrio species on further studies as we need a further fund to have the ability to work on the NGS sequencer.
- Vibrio strain-2 was grouped with different Vibrio spp. including V. natriegens (line 363), Where exactly is the position of isolate NO 2 in the Phylogenetic Tree?
We have added the resolved phylogenetic tree including Vibrio isolate No. 2.
- The treatment of infected shrimp with florfenicol significantly restored the total and differential hemocyte cell counts to their normal levels, relative to the infected group (line 390-391). What about the role or contribution of the florfenicol-resistant beneficial microbes in shrimp immune system stimulation?
Florfenicol was used in a field treatment trial at lower dose of 5mg/kg BWT to restore and improve the survival of shrimp and increase the resistance of vibrio spp. in the treated raceways as the mortality is reduced after treatment. As well, restoring immunity of shrimp was due to increase its resistance to infection via Florfenicol administration, and our priority is to test the efficiency of florfenicol against vibriosis infections not to enhance or stimulate the immunity.

Reviewer 3 Report
MDPI
Manuscript: Fishes-2174464
In this manuscript, the authors studied compsrison of immune response of Litopenaeus vannamei shrimp naturally infected with Vibrio species, and after being fed with florfenicol. The authors presented sequence of primer pairs (Table 1), hatchery field cases of infected and normal broodstook (Figure 2), PCR amplification of the 16S rRNA gene of Vibrio spp. (Figure 3), phylogenetic tree of four Vibrio strains (Figure 4), susceptibility of two pathogenic Vibrio strains to antibiotics (Table 2), total hemocyte count, granular cell, semi-granular cell, hyaline cell (Figure 5), alanine aminotransferase, aspartate aminotransferase, alkaline phosphatase, acid phosphatase (Figure 6), lysozyme, superoxide dismutase, glutathione (Figure 7), phagocytosis, respiratory burst, bactericidal activity (Figure 8), relative qPCR expression of proPO, hemocyanin, LvToll (Figure 9). The authors concluded and reported that florfenicol in medicated feed could be effective in the control of vibriosis and ameliorating the immune response of shrimp. The authors have provided interesting information in the fields of immune response, Vibrio species, and florfenicol.
1. Page 3, 2.1. Collection of shrimp: Change ppt to ‰, and Change to L. vannamei to L. vannamei, change kg L-1 to kg L-1.
2. Page 4, 2.2 Isolation and identification: Change to 28 ± 2 ℃.
3. Page 6, 2.6.2 Immune enzyme and antioxidants: Change to Alkaline phosphatase (ALP and acid phosphatase (ACP).
4. Page 7, 2.7.1: Change to Phagocytosis (%).
5. Page 8, Table 1: Change words in the Reference like Wang et al. (2008), (Soto-Alcalia et al., 2019), (Aguilera-Rivera et al., 2019), (Wang et al.2012a).
6. Page 9, 3.2: Change to (Figure 3).
7. Table 2: Italicize the scientific names.
8. Page 13, Figure 5: Change to Total hemocyte count (THC) and delete counts.
9. Page 16, Figure 8: Change to Phagocytosis (%).
10. Page 16, Figure 9: Change to LvproPO, LvHC, LvToll1
11. References: Check and write full name of Journal, and write italicize scientific names. Also check Fish & shellfish immunology, and Fish & Shellfish Immunology.
Author Response
Manuscript: Fishes-2174464
In this manuscript, the authors studied compsrison of immune response of Litopenaeus vannamei shrimp naturally infected with Vibrio species, and after being fed with florfenicol. The authors presented sequence of primer pairs (Table 1), hatchery field cases of infected and normal broodstook (Figure 2), PCR amplification of the 16S rRNA gene of Vibrio spp. (Figure 3), phylogenetic tree of four Vibrio strains (Figure 4), susceptibility of two pathogenic Vibrio strains to antibiotics (Table 2), total hemocyte count, granular cell, semi-granular cell, hyaline cell (Figure 5), alanine aminotransferase, aspartate aminotransferase, alkaline phosphatase, acid phosphatase (Figure 6), lysozyme, superoxide dismutase, glutathione (Figure 7), phagocytosis, respiratory burst, bactericidal activity (Figure 8), relative qPCR expression of proPO, hemocyanin, LvToll (Figure 9). The authors concluded and reported that florfenicol in medicated feed could be effective in the control of vibriosis and ameliorating the immune response of shrimp. The authors have provided interesting information in the fields of immune response, Vibrio species, and florfenicol.
- Page 3, 2.1. Collection of shrimp: Change ppt to ‰, and Change to L. vannamei to L. vannamei, change kg L-1 to kg L-1.
According to your kind suggestion, we changed these parts.
- Page 4, 2.2 Isolation and identification: Change to 28 ± 2 ℃.
We have modified it.
- Page 6, 2.6.2 Immune enzyme and antioxidants: Change to Alkaline phosphatase (ALP and acid phosphatase (ACP).
We changed it.
- Page 7, 2.7.1: Change to Phagocytosis (%).
We changed it.
- Page 8, Table 1: Change words in the Reference like Wang et al. (2008), (Soto-Alcalia et al., 2019), (Aguilera-Rivera et al., 2019), (Wang et al.2012a).
We have modified it.
- Page 9, 3.2: Change to (Figure 3).
We have modified it.
- Table 2: Italicize the scientific names.
We have modified it.
- Page 13, Figure 5: Change to Total hemocyte count (THC) and delete counts.
We have modified it.
- Page 16, Figure 8: Change to Phagocytosis (%).
We have modified it.
- Page 16, Figure 9: Change to LvproPO, LvHC, LvToll1
We have modified it.
- References: Check and write full name of Journal, and write italicize scientific names. Also check Fish & shellfish immunology, and Fish & Shellfish Immunology.

Round 2
Reviewer 1 Report
The authors did a great job considering my suggestions.